# TRAINING-FREE DEEP CONCEPT INJECTION ENABLES LANGUAGE MODELS FOR CROSSMODAL TASKS

## ABSTRACT

Recently, enabling pretrained language models (PLMs) to perform zero-shot crossmodal tasks has been extensively studied. A popular approach is to learn a projection network that projects visual features into the input text embedding space of a PLM, as well as feed-forward adaptation layers, with the weights of the PLM frozen. However, is it really necessary to learn such additional layers? In this paper, we make the first attempt to demonstrate that the PLM is able to perform zero-shot crossmodal tasks without any training, when the observed visual concepts are injected as both additional input text tokens and augmentation in the intermediate features within each feed-forward network for the PLM. Specifically, inputting observed visual concepts as text tokens helps to inject them through the self-attention layers in the PLM; to augment the intermediate features in a way that is compatible with the PLM, we propose to construct adaptation layers based on the intermediate representation of concepts (obtained by solely inputting them to the PLM). These two complementary injection mechanisms form the proposed Deep Concept Injection, which comprehensively enables the PLM to perceive instantly as learning process is no longer needed. Extensive empirical analysis on zero-shot video question answering and visual question answering shows Deep Concept Injection achieves competitive or even better results, compared to state-of-the-art methods requires crossmodal training.

## 1 INTRODUCTION

Pretrained language models (PLMs) have been shown to be a powerful base model to deal with tasks beyond natural language processing, such as visual question answering (Lu et al., 2019; Dai et al., 2022) and video question answering (Sun et al., 2019; Li et al., 2020a; Lin et al., 2021; Yang et al., 2021; 2022b). These tasks require reasoning over information from multiple modalities and thus the key challenge is to find a common representation so that the information from different modalities can be fused and processed by the PLM. Conventional methods (Lu et al., 2019; Sun et al., 2019) usually rely on a two-stage training process to obtain satisfying results on downstream datasets. Assuming pretrained language models and feature extractors like vision-text contrastive models (e.g., CLIP (Radford et al., 2021)) are available, the first stage aims at crossmodal pretraining on webly-collected vision-text dataset with techniques like masked token modeling (Li et al., 2020a; Zellers et al., 2021) or contrastive learning (Xu et al., 2021; Li et al., 2022; Yang et al., 2021) to learn the alignment and fusion of visual and textual inputs. In the second stage, the model is further fine-tuned with human annotation on specific downstream datasets (Antol et al., 2015; Yang et al., 2021; Yu et al., 2019; Li et al., 2020a; Xu et al., 2017; Lei et al., 2018; Marino et al., 2019) to obtain better models for specific tasks.

However, such a two-stage training process has been criticized to be lack of efficiency, flexibility and generalization (Lin et al., 2021; 2022; Yang et al., 2022b; Li et al., 2023). Therefore, recently researchers (Yang et al., 2022b; Li et al., 2023) have been actively exploring the possibility of relying solely on the first crossmodal pretraining stage and aims at learning a general vision-language model that can perform well without any additional downstream fine-tuning. Successful representative methods in this line of work like FrozenBiLM (Yang et al., 2022b) and BLIP2 (Li et al., 2023) freeze the language model and only train a few projection layers (as well as a few adaptation layers) during the training process to improve the efficiency. This line of research, while notable

for its effectiveness, raises a pertinent question: Is the training of such projection networks truly a necessity?

In this paper, we challenge the prevailing methodology and propose an alternative method that eliminates the need for training projection networks while enabling the PLMs to perform zero-shot crossmodal tasks. As in Figure 1, our approach, Deep Concept Injection (DCI), injects the observed visual concepts as both additional input text tokens and augmentation in intermediate features within each feed-forwards network to enable PLMs to perceive and reason over multimodal inputs.

Our key insights are two-fold. First, towards zero-shot crossmodal tasks, it is necessary to represent the observed visual information in a way that the PLM directly understands, and our solution is to represent the observation using concepts. Inspired by Lin et al. (2022) and Wang et al. (2022), these visual concepts can be extracted through retrieval over a predefined vocabulary given the visual input, with the help of pretrained vision-text contrasting models like CLIP (Radford et al., 2021).

Second and more importantly, in modern PLMs based on Transformers (Vaswani et al., 2017), there are two complementary ways of fusing multimodal information. One commonly used way is to provide visual information as additional elements in the input, where the interaction between visual input and textual input is modeled in the self-attention layers. *However, self-attention layers were trained on natural sentences but not between concept words and a natural sentence. Moreover, the other possibility within feed-forward networks has been ignored.* We propose to leverage the intermediate representations of concept words (when they are solely input to the PLM) to construct adaptation layers and to achieve crossmodal fusion by estimating conditional distribution of the concept given the visual observation and the current word being processed in the PLM.

With the above two key insights, there remains one design choice to complete Deep Concept Injection: how do we choose the set of concepts? One intuitive solution is to leverage existing ontology in computer vision datasets (Krizhevsky et al., 2012; Krishna et al.,

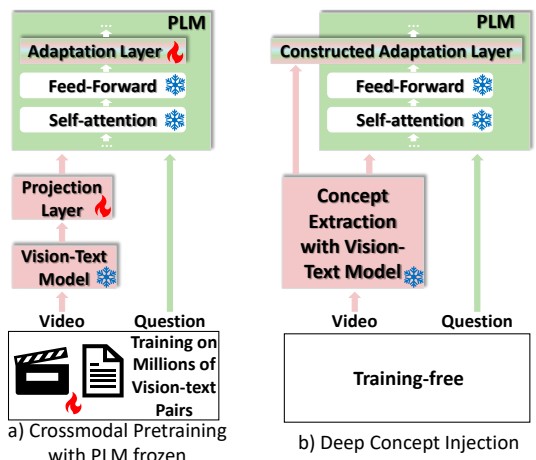

Figure 1: Unlike existing methods of crossmodal pretraining on millions of vision-text pairs, our Deep Concept Injection enables PLMs for zero-shot crossmodal tasks in a training-free manner. The core idea is to leverage concepts as the bridge to inject the visual information in the inference process of PLMs as both input and constructed adaptation layers.

2017; Carreira & Zisserman, 2017). However, such generic datasets might not be aligned with the specific downstream tasks we are interested in. To obtain task-relevant prior, we explore two orthogonal solutions. We first exploit the setting where the access to all the possible answer words of the dataset is allowed, which is actually true for open-ended question answering datasets (Xu et al., 2017; Yu et al., 2019; Yang et al., 2021). Second, to further eliminate the assumption over prior information about the task and dataset, we propose to obtain the set of relevant concepts by querying the language model. With extensive empirical analysis on eleven datasets, the proposed Deep Concept Injection achieves competitive or even better performance than state-of-the-art methods, without any additional training. We believe this paper will stimulate further research and exploration in the field, potentially opening new paths towards more efficient and versatile utilization of PLMs for crossmodal tasks.

The contribution of this paper could be summarized three-fold:

- We first challenge the current methodology of zero-shot crossmodal tasks on the necessity of training additional layers and provide a negative answer by injecting observed visual concepts to PLMs to enable zero-shot crossmodal tasks without any additional training;

- We propose Deep Concept Injection to introduce visual information to PLMs by both inputting the most probable concepts as additional textual input and constructing adaptation layers conditioned observed concepts, which comprehensively enables interaction and fu-

sion of visual and text inputs in both commonly-used self-attention layers and previously-ignored feed-forward layers;

- We provide extensive empirical analysis to facilitate future research, including but not limited to the effect of what prior information to use to narrow down the vocabulary, comparisons with other baselines that don't require additional training and the effect of applying Deep Concept Injection to state-of-the-art models (in the appendix).

## 2 RELATED WORK

**Pre-trained Vision-Text Contrastive Models.** Recently, a family of contrastively pre-trained models are introduced, which are learned from large-scale vision-text data (Miech et al., 2020; Radford et al., 2021; Li et al., 2023). These models typically contain a visual encoder and a text encoder, and learn to map visual and text embeddings into a common space. They sample positive/ negative pairs from aligned/unaligned image/video and texts, and train the visual and text encoders with a contrastive objective in a self-supervised manner. With access to large-scale multimodal data (e.g., 400 million web image-text pairs), they are shown superior on zero-shot recognition tasks. The resulted visual encoders have been also shown to be great feature extractors for downstream tasks (Li et al., 2020b; Yang et al., 2021; 2022b; Wang et al., 2022; Shen et al., 2021; Zhang et al., 2022).

**Crossmodal Tasks with Pretrained Language Models.** Conventional methods (Lu et al., 2019; Sun et al., 2019; Yang et al., 2021) usually rely on a two-stage training process to obtain satisfying results on downstream datasets. Assuming pretrained language models and feature extractors like vision-text contrastive models (e.g., S3D (Miech et al., 2020) and CLIP (Radford et al., 2021)) are available, the first stage aims at training on webly-collected vision-text dataset with techniques like masked token modeling (Li et al., 2020a; Zellers et al., 2021) or contrastive learning (Xu et al., 2021; Luo et al., 2021; Li et al., 2022; Yang et al., 2021) to learn to align and fuse visual and textual inputs. In the second stage, the model is further fine-tuned with human annotation on specific downstream datasets (Yang et al., 2021; Yu et al., 2019; Li et al., 2020a; Xu et al., 2017; Zhou et al., 2018; Wang et al., 2019) to obtain better models for the specific tasks.

Such a two-stage training process has been criticized to be lack of efficiency and flexibility because of the huge cost of the first training stage (Lin et al., 2021; 2022), and they are also not general enough (Yang et al., 2022b; Li et al., 2023). There are two lines of following research trying to address the limitation of the two-stage training process. One line of work (Lin et al., 2021; 2022) focuses on obtaining competitive models with only the second training stage on downstream datasets and one successful idea is to transform every modality into concept text (Lin et al., 2021; 2022) so that the PLM can immediately understand and leverage the information from other modalities without the expensive first training stage. However, such methods still rely on human annotation and need to be trained specifically towards each downstream dataset.

The other line of work (Alayrac et al., 2022; Yang et al., 2022b; Li et al., 2023) relies solely on the first training stage and aims at learning a general vision-language model that can perform well in the zero-shot setting without any additional downstream fine-tuning. During the training process, successful methods in this line of work like FrozenBiLM (Yang et al., 2022b) and BLIP-2 (Li et al., 2023) freeze the language model and only train a few projection layers (as well as a few feed-forward adaptation layers in FrozenBiLM) to project the visual features extracted by a frozen feature extractor like CLIP, to improve the efficiency. The typical training target is, with the video/image as input, generating the associated text. Unlike either of the two lines, we explore a more challenging new problem where there is no additional training or labeled training samples for downstream tasks.

## 3 TECHNICAL APPROACH

In this section, we first present some preliminaries and then introduce the Deep Concept Injection in detail. We propose DCI based on two key ideas: speak the "language" that PLMs understand and comprehensively leverage both ways in Transformer block for crossmodal fusion. The first idea motivates us to leverage concepts (e.g., action, objects, attributes and etc.) as the bridge to transform visual information into text representations. The second idea motivates us to utilize both

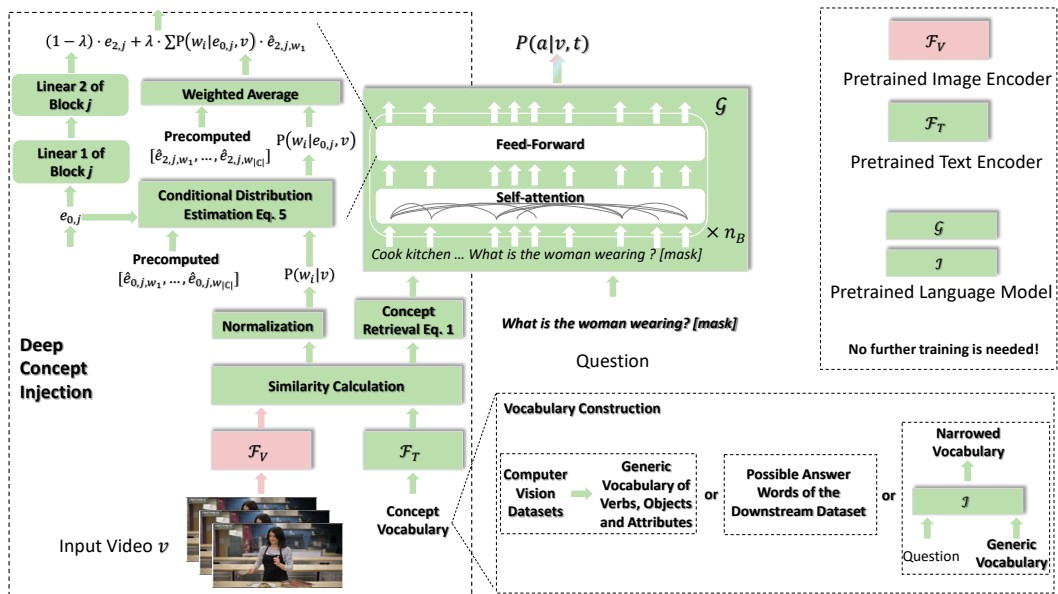

Figure 2: Injecting the observed visual concepts as both additional input text tokens and augmentation in the intermediate features within each feed-forwards network for the PLM enables zero-shot crossmodal tasks without any further training. The most probable concepts extracted from visual input are additional input text so that visual information will be fused with textual information in the self-attention layers (intuitively, "cook, kitchen, ..." provide context for the question); the concept information is further injected in every feed-forward network via adding intermediate representation of concepts weighted with the conditional distribution given current word being processed and the visual input (intuitively, "cook, kitchen, ..." + "wearing" makes it closer to "apron"). Detailed descriptions of the proposed Deep Concept Injection could be found in Sec 3. This figure is best viewed in color when zoomed in.

self-attention layers and feed-forward networks for crossmodal fusion. Last we discuss possible ways of acquiring prior information for vocabulary construction.

## 3.1 PRELIMINARIES

**Crossmodal tasks.** These tasks require the model to fuse information from multiple modalities, e.g., vision and text to return a text response. Specifically, we mainly consider video question answering and visual question answering tasks in this paper. In video question answering, given a video $v$ and question $t$ as input, the model is required to predict the correct answer that matches the ground-truth $a_l$ from an answer corpus $\mathbb{A} = \{a_1, .., a_{|\mathbb{A}|}\}$. In visual question answering, the problem setting is identical and the only difference is that the visual input is a single image. In the model descriptions, we will adopt video question answering for illustration.

**Pretrained Vision-Text Contrastive Models.** We mainly leverage pretrained image-text contrastive models. It consists of a visual encoder $\mathcal{F}_V : \mathbb{R}^{H \times W} \longrightarrow \mathbb{R}^D$ and a text encoder $\mathcal{F}_T : \mathbb{W}^L \longrightarrow \mathbb{R}^D$, where $H, W$ are the height and width, $L$ is the length of the sentence, $D$ is the dimension of the common embedding space and $\mathbb{W}$ is the set of all the words. In this paper, we mainly use it as the concept extractor because of its strong zero-shot recognition abilities (Radford et al., 2021).

**Pretrained Language Models.** The key is to train a model $\mathcal{G} : \mathbb{W}^L \longrightarrow \mathbb{R}^{|\mathbb{W}|}$ that predicts the probability of a word given certain context as input. Depending on the actual objective design, the prediction could be for a masked word (Devlin et al., 2018; He et al., 2020) or the next word (Raffel et al., 2019; Chung et al., 2022). The network architecture could be also categorized as encoder-only (Devlin et al., 2018; He et al., 2020), encoder-decoder (Raffel et al., 2019; Chung et al., 2022), or decoder-only (Brown et al., 2020). All the PLMs used in this paper are based on Transformer (Vaswani et al., 2017), which consists of $n_B$ Transformer blocks and each block's main building components are self-attention layers that models the interaction among different words, and

feed-forward networks that process each word individually. The feed-forward network essentially consists of two linear layers with one activation layer in-between.

## 3.2 Deep Concept Injection

In this section, we describe how to inject observed concepts comprehensively and enable crossmodal fusion in both self-attention layers and feed-forward networks.

### 3.2.1 Injection as Additional Textual Input

To enable crossmodal fusion through self-attention, we extract visual concepts as additional textual input through the retrieval process as follows. First, we construct the word vectors from a predefined concept vocabulary $\mathbb{C}$; specifically, for each word $c_i$, we use the text encoder to obtain its word vector $\mathcal{F}_T(w_i)$. For the input video $v$, we encode it with the pretrained image encoder $\mathcal{F}_V(v)$ frame by frame. Then we compare the similarity between the frame embeddings and each of the words to retrieve $k$ most similar words,

$$w_{1,1}, ..., w_{1,k}, w_{2,1}, ..., w_{F,k} = \arg\max_i^k \mathcal{F}_T(w_i)^\top \mathcal{F}_V(v), \tag{1}$$

where $F$ is the number of frames in the video $v$.

Then the retrieved concepts are fed into the pretrained text model with the question $t$ in parallel to obtain final prediction about answer $a_l$,

$$P(a_l|v,t) = \mathcal{G}(w_{1,1}, ..., w_{1,k}, w_{2,1}, ..., w_{F,k}, t). \tag{2}$$

We follow the temporal order of frames to concatenate retrieved words frame by frame with the question sentence $t$. Note for simplicity, we use a single variable $t$ to denote the actual sentence of the question and the context text, which contains multiple words. As shown in Figure 2, "cook, kitchen, ..." will interact with question words in the self-attention layer and help to provide information about visual observation, which helps the model to reason over multimodal inputs.

### 3.2.2 Injection as Augmentation in the Intermediate Features of Feed-Forward Networks

Since the concept words are not really natural sentences and thus the interaction is not perfectly modeled in the self-attention layers. The ignored possibility of mutlimodal fusion in PLMs lies in the feed-forward networks. We first describe how the augmentation can be added in a way that the PLM understands and then describe why this process can be considered as constructing adaptation layers.

The key of realizing any training-free augmentation for a pretrained model is to speak in the "language" that the model understands. Therefore, we first extract intermediate representation of each concept when they are input to the PLM individually,

$$\hat{e}_{0,j,w_i} = \mathcal{G}_{0,j}(w_i), \tag{3}$$

where $\hat{e}_{0,j,w_i}$ represents the intermediate representation of a concept $w_i$, which is input to the feed-forward network in the $j$-th Transformer block of the PLM. Similarly, we can extract the output representation of the feed-forward network in each Transformer block for each concept word,

$$\hat{e}_{2,j,w_i} = \mathcal{G}_{2,j}(w_i). \tag{4}$$

Note that these extraction processes only need to be done once for all the future crossmodal inference, which makes the amortized complexity to be negligible.

As shown in Figure 2, during inference for crossmodal tasks as in Eq. 2, for simplicity, we denote the input intermediate representation and the output intermediate representation of whichever word is currently being processed as $e_{0,j}$ and $e_{2,j}$, respectively. To fuse crossmodal information, we first compute the conditional distribution with the approximation that $e_{0,j}$ is independent of $v$,

$$P(w_i|e_{0,j}, v) \approx \frac{P(w_i|e_{0,j})P(w_i|v)}{P(w_i)}. \tag{5}$$

The factorized terms can be obtained as follows,

$$P(w_i|e_{0,j}) = \frac{\exp\left(\hat{e}_{0,j,w_i}^\top e_{0,j}\right)}{\sum_l \exp(\hat{e}_{0,j,w_l}^\top e_{0,j})}, \tag{6}$$

$$P(w_i|v) = \text{Top}_k(\text{Max-pool}(\frac{\exp\left(\mathcal{F}_T(w_i)^\top \mathcal{F}_V(v)\right)}{\sum_l \exp(\mathcal{F}_T(w_l)^\top \mathcal{F}_V(v))})), \tag{7}$$

where the Max-pool is applied along the temporal axis for the video input to handle multiple input frames and $\text{Top}_k$ indicates that we only keep the most relevant $k$ concept's probability to be non-zero and then scale the distribution so that the summation of probabilities is 1. This process essentially keeps the most relevant and probable visual concepts of the visual input, which we also find important empirically. We don't assume extra information about $P(w_i)$ and thus we simply apply the uniform distribution. In practice, we simply scale the product of $P(w_i|e_{0,j})$ and $P(w_i|v)$ to ensure the summation to be 1 to obtain the estimation of $P(w_i|e_{0,j}, v)$.

Then we leverage the conditional distribution to augment the output intermediate representation of the feed-forward network by adding the representation of concepts weighted based on the conditional distribution,

$$e_{2,j} = (1 - \lambda) \cdot e_{2,j} + \lambda \cdot \sum_i P(w_i|e_{0,j}, v) \cdot \hat{e}_{2,j,w_i}. \tag{8}$$

Both the calculation of the conditional probability and the augmentation of the output intermediate representation can be done in parallel for each word as matrix multiplication, which leads to the equivalence to a feed-forward adaptation network

$$e_{2,j} = (1 - \lambda) \cdot e_{2,j} + \lambda \cdot \text{Linear}_2(\text{Act}(\text{Linear}_1(e_{2,j}; \theta_1)); \theta_2), \tag{9}$$

where $\theta_2$ is the weight matrix of the second linear layer $\text{Linear}_2$ whose row $i$ is the transpose of $\hat{e}_{2,j,w_i}$, $\theta_1$ is the weight matrix of the first linear layer $\text{Linear}_1$ whose column $i$ is $\hat{e}_{0,j,w_i}$ and Act consists of both soft-max and element-wise multiplication with $P(w_i|v)$.

Intuitively, as verified in Figure 4, intermediate representation of "[mask]" could not be close to the answer "hat" but after adding the representation of observed concepts, the model can make correct prediction. Therefore, by further injecting the visual concept in the feed-forward network of each block, the visual information is comprehensively fused with the textual input for the PLM to make better prediction for crossmodal tasks.

### 3.3 PRIOR INFORMATION ACQUISITION FOR VOCABULARY CONSTRUCTION

Existing computer vision datasets provide a generic vocabulary of visual concepts $\mathbb{C}$. Inspired by (Wang et al., 2022), we curate a comprehensive visual concept vocabulary of verbs, objects and attributes from Visual-Genome (Krishna et al., 2017; Kuznetsova et al., 2020). We denote the variant using this generic vocabulary as DCI. However, such a vocabulary could be too general for downstream tasks.

We first explore a setting with the access to the answer word vocabulary which either consists of the most frequent answers from the training set provided in the open-ended setting or consists of the answer words from the choices in the multiple-choice setting. This does not leak any information for 8 datasets of open-ended video question answering. We denote this variant as DCI-A.

To generally obtain prior information about the task to narrow down from a generic vocabulary, we propose to prompt a PLM to ask about relevant visual concepts

$$P(w_i|I) = \mathcal{I}(t), \tag{10}$$

where $t$ is the question (and context) and $\mathcal{I}$ is not necessarily the same PLM we use for crossmodal tasks, although in our implementation we use the same model for simplicity of implementation. Then we can narrow down a subset of most $n_c$ probable concept words from the generic vocabulary $\mathbb{C}$. We denote this variant as DCI-LM.

Table 1: Comparison with the zero-shot state-of-the-art on video question answering in terms of accuracy (%) and efficiency. Our DCI is built upon CLIP and DeBERTa-V2, as FrozenBiLM. MM Samples indicate the number of video-text samples used in the crossmodal pretraining process. GPU hours denote the additional computation required for it. Bold indicates the best results. "-" is not applicable and "?" means unclear from the original paper.

| Model | MM Samples | GPU hours | iVQA | ActivityNet-QA | TGIF-QA | How2QA | TVQA | LSMDC | MSRVTT-QA | MSVD-QA |
|---|---|---|---|---|---|---|---|---|---|---|
| Random | - | - | 0.1 | 0.1 | 0.1 | 25.0 | 20.0 | 0.1 | 0.1 | 0.1 |
| VQA-T (Yang et al., 2022a) | 69M + 3M | 350 + 30 | 13.3 | 12.3 | - | 53.1 | - | - | 5.6 | 13.5 |
| Reserve (Zellers et al., 2022) | 1B | 196K | - | - | - | - | - | 31.0 | 5.8 | - |
| Flamingo3B (Alayrac et al., 2022) | 2.1B | ? | 32.7 | - | - | - | - | - | - | 27.5 |
| Flamingo9B (Alayrac et al., 2022) | 2.1B | ? | 35.2 | - | - | - | - | - | - | 30.2 |
| Flamingo80B (Alayrac et al., 2022) | 2.1B | 553K | **40.7** | - | - | - | - | - | - | **35.6** |
| CLIP (Radford et al., 2021) | - | - | 9.2 | 1.2 | 3.6 | 47.7 | 26.1 | 1.2 | 2.1 | 7.2 |
| DeBERTa-V2 (He et al., 2020) | - | - | 12.1 | 23.0 | 32.3 | 52.7 | 55.1 | 50.0 | 6.5 | 11.7 |
| FrozenBiLM (Yang et al., 2022b) | 10M | 160 | 26.8 | 25.9 | 41.9 | 58.4 | 59.7 | 51.5 | 16.7 | 33.8 |
| DCI (ours) | 0 | 0 | 28.0 | 25.5 | 45.2 | 58.7 | 60.4 | 51.7 | 17.2 | 34.5 |
| **DCI-A (ours)** | 0 | 0 | 30.2 | **26.0** | **45.6** | 59.0 | 59.9 | **52.2** | **17.6** | 35.1 |
| **DCI-LM (ours)** | 0 | 0 | 28.5 | 25.9 | 45.3 | **59.4** | **60.5** | 52.1 | 17.4 | 34.4 |

# 4 EXPERIMENTAL RESULTS

In this section, we will first introduce the implementation and evaluation settings. Then we organize the following subsections by answering a set of important questions. More ablations, further analysis and other details are provided in the appendix.

## 4.1 IMPLEMENTATION AND EVALUATION SETTINGS.

We mainly compare with two state-of-the-art models using frozen PLMs and learned projection layers, FrozenBiLM and BLIP-2. We follow their settings respectively to implement and evaluate our methods. Based on experiments in the appendix, we use $k = 4$, $\lambda = 0.01$, and $n_c = 1500$.

FrozenBiLM is evaluated on 8 video question answering datasets: iVQA (Yang et al., 2021), ActivityNet-QA (Yu et al., 2019), TGIF-QA (Jang et al., 2017), How2QA (Li et al., 2020a), TVQA (Lei et al., 2018), LSMDC (Maharaj et al., 2017), which are manually labeled; MSRVTT-QA (Xu et al., 2017) and MSVD-QA (Xu et al., 2017), which are generated automatically from video captions. We follow its evaluation setting for each of the datasets to report results. Our models use the same CLIP ViT-L/14 (Radford et al., 2021) model and the same DeBETa-V2-XL (He et al., 2020) model as the FrozenBiLM model.

BLIP-2 is evaluated on VQAv2 (Goyal et al., 2017), OK-VQA (Marino et al., 2019), and GQA (Hudson & Manning, 2019). We use the same evaluation setting for each of the datasets as BLIP-2. We use the same pretrained Q-Former based on ViT-g (Fang et al., 2022) and the pretrained FlanT5-XXL (Chung et al., 2022). Because of the Q-former, the extracted features of an image will have an axis for different learned queries, which can be handled in the same way as the temporal dimension in the video question answering setting illustrated in Section 3.

## 4.2 HOW DOES DCI COMPARE WITH STATE-OF-THE-ART METHODS THAT TRAIN PROJECTION LAYERS (AND ADAPTATION LAYERS)?

As shown in Table 1, compared to state-of-the-art zero-shot video question answering model Frozen-BiLM, without training on 10 million video-text pairs for 160 GPU hours, all the proposed DCI variants generally achieve better or competitive results on all the 8 video question answering datasets. On some of the datasets like iVQA and TGIF-QA, the absolute improvement is up to 3.7% and the relative improvement is up to 12.7%. In spite of the huge difference in terms of the number of parameters in the model (890M v.s. 80B) and the huge number of multimodal samples (2.1B) and cost of training (553K TPU hours), compared to Flamingo80B, our proposed DCI method based on DeBERTa-v2-XL can still achieve comparable performance on some of the datasets like MSVD-QA.

As shown in Table 2, compared to state-of-the-art zero-shot visual question answering model BLIP-2, without training on 129 million video-text pairs for 1 thousand GPU hours, all the proposed DCI variants still generally achieve better or competitive results on all the 3 visual question answering datasets. It is noteworthy that on VQAv2, with a smaller PLM FlanT5-XXL (12B), the proposed DCI

Table 2: Comparison with the zero-shot state-of-the-art on visual question answering in terms of accuracy (%) and efficiency. Our DCI is built upon the same pretrained models as BLIP-2 ViT-g FlanT5XXL. MM Samples indicate the number of image-text samples used in the crossmodal pretraining process. GPU hours refer to the additional computation required for it. Bold indicates the best results. "-" means not applicable and "?" means unclear from the original paper.

| Model | MM Samples | GPU hours | VQAv2 test-dev | OK-VQA test | GQA test-dev |
|---|---|---|---|---|---|
| VLKD (Dai et al., 2022) | 3.7M | 320 | 44.5 | 13.3 | - |
| Flamingo3B (Alayrac et al., 2022) | 2.1B | ? | 49.2 | 41.2 | - |
| Flamingo9B (Alayrac et al., 2022) | 2.1B | ? | 51.8 | 44.7 | - |
| Flamingo80B (Alayrac et al., 2022) | 2.1B | 553K | 56.3 | **50.6** | - |
| BLIP-2 (Li et al., 2023) | 129M | 1K | 65.0 | 45.9 | 44.7 |
| DCI (ours) | 0 | 0 | 64.5 | 46.3 | 45.2 |
| **DCI-A (ours)** | 0 | 0 | **65.9** | 46.8 | **45.4** |
| **DCI-LM (ours)** | 0 | 0 | 65.4 | 46.9 | 45.2 |

Table 3: Comparison between FrozenBiLM and its counterpart without training. "Projection Layer" indicates how the projection layers are obtained. * denotes no adaptation layers are used.

| Model | Projection Layer | iVQA | ActivityNet-QA | TGIF-QA | How2QA | TVQA | LSMDC | MSRVTT-QA | MSVD-QA |
|---|---|---|---|---|---|---|---|---|---|
| FrozenBiLM | Learned | 26.8 | 25.9 | 41.9 | 58.4 | 59.7 | 51.5 | 16.7 | 33.8 |
| FrozenBiLM* | Learned | **27.3** | 24.7 | **41.0** | 53.5 | 53.4 | 50.7 | **16.8** | 32.2 |
| CLIP+DeBERTa | Random | 7.0 | 14.2 | 22.8 | 46.8 | 39.4 | 46.8 | 4.3 | 7.1 |
| CLIP+DeBERTa | Constructed | 24.5 | 24.1 | 39.5 | 55.8 | 57.9 | 51.0 | 15.9 | 32.6 |
| CLIP+DeBERTa | Concepts | 26.5 | **25.1** | 40.8 | **57.6** | **59.4** | **51.4** | 16.7 | **33.5** |

even outperforms Flamingo80B by 9.6% of absolute accuracy. All these results are very encouraging and surprising, which provide a concrete negative answer on the necessity of training projection networks (FrozenBiLM and BLIP-2) and even a few lightweight adaptation layers (FrozenBiLM) for zero-shot crossmodal tasks.

### 4.3 HOW DOES DIFFERENT VOCABULARY CONSTRUCTION METHODS EFFECT ZERO-SHOT MUTLIMODAL REASONING PERFORMANCE?

As shown in Table 1 and 2, we observe that generally the DCI-A variant performs the best, which is expected as the possible answer words in each dataset provide strong prior information about the task and the dataset. We also find that using the PLM to narrow down from the generic vocabulary always helps to improve the performance but not as significant as DCI-A. As the hyper-parameters are tuned with only iVQA validation set, it is still encouraging to observe a rather consistent improvement from DCI-LM.

### 4.4 ARE THERE OTHER WAYS OF ZERO-SHOT CROSSMODAL TASKS WITHOUT ANY ADDITIONAL TRAINING?

Based on the insights discussed in Eq. 9, we provide a baseline with a constructed projection layer that requires no additional training and also helps us understand methods like FrozenBiLM. The main idea is instead of learning the projection layers, the "projected" visual features in the text embedding space could be obtained by weighted-averaging concept embeddings with the conditional distribution of concepts given the visual input. Formally, $e_t = \sum_i P(w_i|v)_t \cdot e_{w_i}$, where $e_t$ is the "projected" visual feature of the $t$-th frame and $e_{w_i}$ is the word embedding of word $w_i$. We further provide another baseline where instead of weighting the word embeddings of concepts, we directly concatenate the most relevant concepts as additional textual input. This baseline is essentially only injecting concepts as inputs, without augmentation in the intermediate features.

As in Table 3, we comprehensively evaluate these baselines on 8 video question answering datasets and this baseline performs surprisingly well. The constructed variant significantly outperforms the random initialization and performs slightly lower than the learned FrozenBiLM, which indicates that most of the ability of the learned projection layers and the adaptation layers can be instantly obtained with the simple constructed projection layer. Such constructed projection layers or learned projection layers are inferior to directly appending the most relevant concepts, which indicates that a sequence of concept words is better representation than a single projected visual feature.

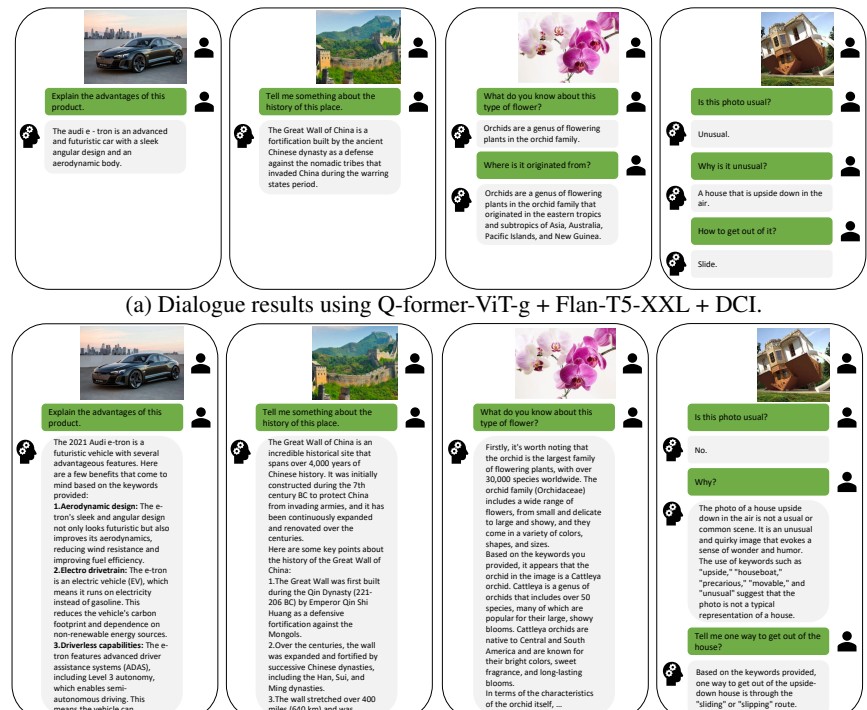

(a) Dialogue results using Q-former-ViT-g + Flan-T5-XXL + DCI.

(b) Dialogue results using Q-former-ViT-g + LLAMA2-7B-Chat + DCI.

Figure 3: The proposed DCI method generalizes well to multimodal dialogues with different PLMs.

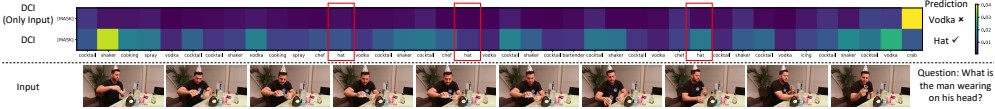

Figure 4: Attention visualization of DCI with only injections as inputs and full DCI. With the help of augmentation in the intermediate features, "[mask]" token attends more to "hat", which leads to the correct prediction. Best viewed when zoomed in.

## 4.5 QUALITATIVE RESULTS

**Zero-shot Multimodal Dialogue.** We show the zero-shot dialogue results in Figure 3 of the provided PDF. We find the zero-shot multimodal dialogue results to be pretty impressive. With the proposed DCI method, PLMs such as FLAN-T5-XXL and the latest LLAMA2-7B-Chat can instantly be used for multimodal dialogue without any training. For example, the PLMs can successfully understand the input image containing an upside-down house, and address reasoning questions like how to get out of the house based on visual information and dialogue context.

**Attention Visualization.** In Figure 4, we visualize the average attention in the last transformer block for results from DCI with only injection as inputs and full DCI. We observe that the augmentation in the intermediate feature space helps the model attend more to extracted concepts that are relevant to the correct answer. This again verifies that the proposed two injection mechanisms are complementary to each other.

## 5 CONCLUSION

In this paper, we present a novel approach to enabling pretrained language models to perform zero-shot crossmodal tasks. The proposed Deep Concept Injection, effectively circumvents the necessity of training projection networks, a widely accepted practice in this field, and instead makes insightful use of observed visual concepts as additional input text tokens and as means for augmenting intermediate features. Extensive results show that they function synergistically to realize strong zero-shot crossmodal capabilities of the PLM. We leave the discussion on limitation, future work and broader impact in the appendix.

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

## A  APPENDIX

In the following sections, we will provide:

- Results of plugging DCI into existing trained models (FrozenBiLM);
- Speed comparison;
- Results of using ImageNet Classification Model for Concept Extraction;

- Discussion on limitation, future work and broader impact;
- Four ablation studies;
- Additional details.
- More discussion on zero-shot multimodal dialogue results

Table 4: Results (%) of plugging DCI-A into FrozenBiLM on iVQA, ActivityNet-QA, TGIF-QA, How2QA, TVQA, LSMDC, MSRVTT-QA and MSVD-QA. "Fine-tuned" indicates whether the FrozenBiLM model is further fine-tuned on each downstream datasets. Bold indicates the better results.

| Model | Fine-tuned? | iVQA | ActivityNet-QA | TGIF-QA | How2QA | TVQA | LSMDC | MSRVTT-QA | MSVD-QA |
|---|---|---|---|---|---|---|---|---|---|
| FrozenBiLM (Yang et al., 2022b) | No | 26.8 | 25.9 | 41.9 | 58.4 | 59.7 | 51.5 | 16.7 | 33.8 |
| **FrozenBiLM + DCI-A (ours)** | No | **30.6** | **26.1** | **46.3** | **59.5** | **59.8** | **52.4** | **17.3** | **35.0** |
| FrozenBiLM (Yang et al., 2022b) | Yes | 39.6 | 43.2 | 68.6 | 86.7 | **82.0** | 63.5 | 47.0 | 54.8 |
| **FrozenBiLM + DCI-A (ours)** | Yes | **40.4** | **43.3** | **69.5** | **87.1** | 81.9 | **63.8** | **47.6** | **55.0** |

## B  CAN DCI SERVE AS A PLUG-AND-PLAY AUGMENTATION FOR MODELS REQUIRING ADDITIONAL TRAINING?

The motivation of DCI is to eliminate additional training and to enable PLMs to directly perform crossmodal tasks. Since there are already existing trained models, it is important and interesting to explore the flexibility of the proposed DCI as a plug-and-play augmentation to these trained models. We take FrozenBiLM for this case study as its trained and fine-tuned checkpoints are all released. Specifically, for the input sequence, we append the retrieved visual concepts between the projected visual features and the question text; for the augmentation in the intermediate representations, we perform exactly the same augmentation process for every input token except the projected visual features.

As shown in Table 4, we extensively evaluate both FrozenBiLM trained with video-text pairs and its variants further fine-tuned on each downstream dataset, with the proposed DCI-A as a plug-and-play augmentation. We observe that even when the projection and adaptation layers are well trained or even fine-tuned towards the specific downstream task, our DCI-A can still help to better fuse the visual information with textual information. This again verifies the necessity of injecting observed concepts and the complementarity with existing approaches.

## C  SPEED COMPARISON

As shown in Table 5, we measure the inference speed on a V100 GPU with batch size 1 on the validation set of the iVQA dataset. The running time is shown in the following table. The increase in running time of DCI is rather tolerable compared to other models like FrozenBiLM. The time of one ablation experiment of DCI typically takes about 1 GPU minute. Models like FrozenBiLM also need hyper-parameter search, which is much more expensive.

Table 5: Inference speed comparison.

| Method | Running time (seconds per iteration) |
|---|---|
| FrozenBiLM | $0.0461 \pm 0.0010$ |
| DCI (Ours) | $0.0495 \pm 0.0013$ |

## D  USING IMAGENET CLASSIFICATION MODEL FOR CONCEPT EXTRACTION

To understand whether our model generalizes beyond vision-text contrastive model for concept extraction, we use the same ViT pretrained on ImageNet21k as FrozenBiLM in its Table 14. As shown in Table 6, The superior results of our DCI achieved again verifies it effectiveness of enabling zero-shot multimodal reasoning without training. The performance is lower than using CLIP for concept extraction as expected, which is also observed by (Alayrac et al., 2022) because "our goal is to use the Vision Encoder as a feature extractor for the Flamingo models in order to capture the whole scene and not just the main object".

Table 6: Comparison with FrozenBiLM on the iVQA dataset when ImageNet pretrained model is used as the feature/concept extractor.

| Method | iVQA Accuracy (%) |
|---|---|
| FrozenBiLM | 23.8 |
| DCI (Ours) | 25.3 |

## E  LIMITATION, FUTURE WORK AND BROADER IMPACT

One limitation in this work is that only crossmodal tasks over vision and text is evaluated. Since we have already covered 11 datasets, we leave further exploiting broader combination and tasks as future work. However, the proposed approach is rather generic: as long as there is a concept extractor for modality X, preferably a pretrained X-text contrastive model for the modality X and text, the proposed DCI can be applied instantly. Another limitation of the proposed method is that it certainly adds additional running time during inference because of the extra computation but the main complexity still comes from the inference of the large PLM itself. We also note that in the current evaluations, the size of the PLM used is still rather limited to a rather small scale. Further scaling up the language model is another interesting future work. We also would like to note that we assume there is no access to good captioning models for all the models evaluated. In practice, further augmenting inputs with captions generated by pretrained captioning models could possibly further improve the performance, which is orthogonal to the setting and approaches explored in this paper. Another interesting future research direction is to further develop approaches that can acquire and inject certain prior information into the inference process of a PLM.

While we do not anticipate direct negative social consequences stemming from the work, it is important to note our work relies on pre-trained models which could potentially exhibit certain biases.

## F  ABLATION STUDIES

In this section, we report the results of ablation studies on the validation set of the iVQA dataset.

### F.1  FINE-TUNING SETTING

Despite this paper is focused on zero-shot setting, we are also interested whether the first training stage on webly-collected video-text pairs bring extra benefit on other settings or not. Therefore, we perform a case study on the iVQA dataset. Similar to FrozenBiLM, we freeze the PLM but just update the parameters of the same adapter networks. As illustrated in Table 7, compared to directly fine-tuning FrozenBiLM without the first training stage, our fine-tuning our DCI-A equipped model significantly improves the accuracy, which demonstrates the effectiveness of the proposed method for fusing visual information beyond zero-shot setting. When comparing with FrozenBiLM with 10M of more examples for training, our DCI-A still outperforms it by 1.7% of accuracy, which further indicates the importance of injecting visual information properly through the proposed mechanism even compared to extensive training.

### F.2  HYPER-PARAMETER SELECTION

We first vary the three hyper-parameters introduced in the proposed DCI method, the number of concepts retrieved, the injection weight, and the vocabulary size when we use the PLM to narrow down from the generic vocabulary. As shown in Table 8a, we observe that using $k = 4$ produces the best results and changing number of words around 4 does not change the performance too much. As presented in Table 8b, we find that using a relatively small $\lambda = 0.01$ for injection as augmentation in the intermediate feature works better. When $\lambda$ is significant larger, the performance degrades, which is intuitively understandable as this would change the intermediate representation of the model too much. As shown in Table 8c, we observe that significantly narrowing down the vocabulary by one order of magnitude helps to improve the accuracy but when the vocabulary is too small the performance would also degrade. Overall, we find that within the range we explored, the performance of the method w.r.t. hyper parameters is stable.

Table 7: Results under fine-tuning setting on the iVQA test set. $\Delta$ Multimodal Samples indicate the number of video-text samples that are needed in the the first training stage on webly-collected video-text pairs. $\Delta$ GPU hours refer to the additional computation required for that training stage. Fine-tuning on the iVQA dataset requires 1 GPU hour for reference. All the compared models are fine-tuned on the iVQA dataset. * indicates FrozenBiLM is fine-tuned without loading pretrained checkpoint from the first training stage.

| Model | $\Delta$ Multimodal Samples | $\Delta$ GPU hours | Accuacy (%) |
|---|---|---|---|
| FrozenBiLM (Yang et al., 2022b) | 10M | 160 | 39.6 |
| Text + Text (Lin et al., 2022) | 0 | 0 | 40.2 |
| FrozenBiLM* | 0 | 0 | 31.6 |
| **DCI-A (ours)** | 0 | 0 | **41.3** |

Table 8: Results for hyper-parameter selection on the iVQA validation set.

(c) The vocabulary size.

(a) The number of retrieved concepts.

(b) The injection weight.

| $k$ | Accuracy (%) |
|---|---|
| 2 | 27.9 |
| 4 | 28.1 |
| 6 | 27.3 |

| $\lambda$ | Accuracy (%) |
|---|---|
| 0.005 | 27.8 |
| 0.01 | 28.1 |
| 0.015 | 28.0 |
| 0.1 | 26.5 |

| $n_c$ | Accuracy (%) |
|---|---|
| 500 | 27.6 |
| 1000 | 28.1 |
| 1500 | 28.5 |
| 2000 | 28.4 |
| 2500 | 28.2 |
| 10738 (Full) | 28.1 |

## F.3 CONTRIBUTION OF THE TWO INJECTION MECHANISMS

We would also like to understand the contribution of the two mechanisms we propose to leverage for zero-shot multimodal reasoning. As shown in Table 9, we observe that injecting observed visual concepts as additional textual context contributes the main improvement over the language model only baseline (no injection is used). The augmentation in the intermediate features within feed-forward networks helps to further improve the performance. We think this is expected as the directly injecting as additional textual input leverages the well-trained self-attention layers to fuse information between text and vision and thus it is easier to provide visual information to the PLM. However, this is not complete or perfect as the PLM may be not able to directly well fuse the visual concepts with other textual input well because the visual concepts are not the same as natural sentences. Augmenting the intermediate features helps to further inject visual information explicitly, which complements the previous mechanism by their designs and is verified by the empirical results.

## F.4 PERFORMANCE BREAKDOWN ON ACTIVYTYNET-QA

We report the detailed performance breakdown based on the manually labeled types of QA in the ActivityNet-QA dataset. We observe that there are certain types of questions that our method achieves significant improvement, such as Color, Number and Yes-No. We believe this is because that these important concepts like colors are directly represented in our method compared to using a projected visual feature vector, which makes it easier for the model to obtain the required information for answering the question. Over all the types, all the methods including our method performs poorly on Temporal-related QA, which indicates a possible future direction for further improvement.

Table 9: Accuracy with different combination of injection mechanisms on the iVQA validation set.

| As Additional Input | As Augmentation in Features | Accuracy (%) |
|:---:|:---:|:---:|
| ✗ | ✗ | 12.1 |
| ✓ | ✗ | 26.3 |
| ✗ | ✓ | 13.2 |
| ✓ | ✓ | 28.1 |

Table 10: Results for different types of QA on the ActivityNet-QA test set.

| Model | Motion | Spatial | Temporal | Yes-No | Color | Object | Location | Number | Other | Overall |
|:---|:---:|:---:|:---:|:---:|:---:|:---:|:---:|:---:|:---:|:---:|
| VQA-T (Yang et al., 2021) | 2.3 | 1.1 | 0.3 | 36.3 | 11.3 | 4.1 | 6.5 | 0.2 | 4.7 | 12.3 |
| FrozenBiLM (Yang et al., 2022b) | **12.7** | **6.8** | **1.6** | 53.2 | 16.5 | 17.9 | **18.1** | 26.2 | **25.8** | 25.9 |
| DCI (ours) | 11.0 | 4.8 | 0.8 | 55.2 | 23.2 | **18.6** | 10.2 | 25.7 | 22.3 | 25.5 |
| DCI-A (ours) | 11.3 | 5.8 | 1.3 | 55.3 | **24.7** | 16.5 | 11.2 | **29.6** | 22.0 | **26.0** |
| DCI-LM (ours) | 10.8 | 4.9 | 1.4 | **55.4** | 24.6 | 16.9 | 11.2 | 29.2 | 22.2 | 25.9 |

# G  ADDITIONAL DETAILS

## G.1  IMPLEMENTATION DETAILS

We implement the DCI method using PyTorch and inject our implementation to publicly available code repositories of FrozenBiLM (Yang et al., 2022b) and BLIP-2 (Li et al., 2023), respectively. We use half precision for model parameters to save memory and improve speed during inference. All the experiments on video question answering are done with 4 Nvidia V100-32GB GPUs. Experiments on visual question answering is done with a Nvidia A100-40GB GPU.

For video question answering tasks, we follow the prompt of FrozenBiLM to query the language model with questions and additional input and determine the answer based on the probability obtained for the "[mask]" token. For visual question answering, we follow the same setting of BLIP-2 to generate answers and then compare with the ground-truth.

To construct the vocabulary, we follow VidIL (Wang et al., 2022) to construct vocabulary. There are 2,138 verbs, 6,369 objects and 7,233 attributes curated for the vocabulary. Merging and deduplication results 10,738 unique concept words. We find that directly using all these concept words together as one vocabulary has already helped, so we do not perform further fine-grained processing among different categories of concept words.

When computing the intermediate representations for each concept word, we simply average the representation if there are multiple tokens in the concept word. For fine-tuning experiments, we follow the same hyper-parameters as used in FrozenBiLM. Our code will be made publicly available upon publication.

## G.2  DATASET AND EVALUATION METRIC FOR ABLATION STUDY

**iVQA (Yang et al., 2021)** contains 10,000 instructional videos. Each video is annotated with one question and five corresponding answers. In the official split, there are 6,000, 2,000, and 2,000 videos for training, validation, and testing, respectively. We use the 2,000 videos in the validation set for ablation study in the appendix (when not specified) to avoid fitting the test set through hyper-parameter selection and follow the test split of all the datasets used in FrozenBiLM and BLIP-2 to report results in the main paper. We follow (Yang et al., 2021) to calculate accuracy with five annotations per question.

# H  MORE DISCUSSION ON ZERO-SHOT MULTIMODAL DIALOGUE RESULTS

One interesting aspect of the results here is that the model was able to recognize some named entities. After checking the reconized concepts, we find that some of the entities are indeed part of the

vocabulary like audi e-tron. For the Great Wall image, the recognized concepts include "china", "fortification", and "tourism". The PLM successfully inferred the most famous Great Wall based on these concepts. Currently, we don't intentionally handle named entities in our vocabulary, but this ability can be further integrated if we can also provide a list of named entities that we want the model to recognize, which will be an interesting future research direction.

