# OpenReview forum: "Training-free Deep Concept Injection Enables Language Models for Crossmodal Tasks"
_ICLR.cc/2024/Conference — Submitted to ICLR 2024_

### Official Review · Reviewer_Hqos · 2023-10-29

**Soundness:** 3 good
**Presentation:** 3 good
**Contribution:** 3 good
**Rating:** 6
**Confidence:** 3

**Summary:**

This paper introduces Deep Concept Injection (DCI) as an approach to achieving zero-shot cross-modal tasks without the need for additional training. This work conducts a comprehensive set of experiments and analyses to validate the effectiveness of this approach.

**Strengths:**

The method of constructing projection networks to replace fine-tuning is intriguing. The authors successfully inject observed concepts, facilitating cross-modal fusion in self-attention layers and feed-forward networks.

**Weaknesses:**

The paper claims to be the first to demonstrate the ability of Pre-trained Language Models (PLMs) to perform zero-shot cross-modal tasks without any training. However, there exist similar works, such as Tip-Adapter [1], which should be discussed and compared to provide context and clarify the novelty of this approach.

[1] Zhang, Renrui, et al. "Tip-adapter: Training-free adaption of clip for few-shot classification." European Conference on Computer Vision. Cham: Springer Nature Switzerland, 2022.

**Questions:**

Does the Deep Concept Injection process incorporate additional information? While there is no explicit training involved in the design of this paper, has any training data been indirectly introduced into this process?

---

> ### Author Response · Authors · 2023-11-15
> **Response to Reviewer Hqos**
>
> Dear Reviewer Hqos,
>
> Thank you for your detailed comments and suggestions. We appreciate your recognition of the soundness, contribution, and presentation of our work. We tried our best to address all the concerns and questions, and update the main paper and appendix in the new version. Please let us know if you have any further concerns or questions to discuss.
>
> Best,
>
> Paper 4148 Authors
>
> ---
>
> ### Our position w.r.t. existing work
> - Thanks for the reference. But the provided reference doesn’t change the position of our paper. **Our work is focused on crossmodal tasks that require modality fusion, but the Tip-adapter is designed only for unimodal classification tasks, which is fundamentally different from ours**. We have cited it in the revised main paper.
>
> ---
>
> ### Information from training data
> - For DCI using the genric vocabulary and DCI-LM, there is no information from the training set at all.
> - As we discussed in Sec 3.3, for the variant DCI-A, the most frequent answers from the training set provided in the open-ended setting are used as the vocabulary. We also noted that this does not leak any additional information for 8 datasets of open-ended video question answering since the compared FrozenBiLM also uses the answer vocabulary as part of the input to the model.

---

> > ### Comment · Reviewer_Hqos · 2023-11-23
> >
> > Thanks for your detailed response. I have no other questions.

---

### Official Review · Reviewer_Eqqp · 2023-10-31

**Soundness:** 2 fair
**Presentation:** 2 fair
**Contribution:** 2 fair
**Rating:** 5
**Confidence:** 3

**Summary:**

This paper proposed a method to inject visual concepts into pretrained LLMs without any training for vision-language tasks. The authors leveraged a pre-extracted concept library to aggregate vision-related concepts into feed forward pretrained LLMs with probabilistic weighting. By properly constructing the concept library, the proposed DeepConceptInjection (DCI) model achieve state-of-the-art results on several vqa and video-qa tasks with significant training overhead reduction.

**Strengths:**

The proposed model is training-free, by properly constructing the concept library and weighting the forward features of words in concept library, the resulting DCI can correctly adapt vision-related information into pretrained-LLMs for vision-language tasks. The resulting model achieve state-of-the-art results compared with existing VQA or Video-QA models with similar model scale.

**Weaknesses:**

Despite the good performance and simplicity of the method, this paper seems to be missing several critical key points:
- First, the concept library seems to be extremely important in the whole DCI pipeline, however, how to construct this concept library given different input datasets or domains seems to be too simple for evaluating if this pipeline could be adapted to more general settings. The authors should consider adding more details to this concept library construction process for the reviewers to evaluate the contribution of this paper.

- The overall architecture seems extremely simple and efficient. Given the good performance, the authors should have considered providing more insights on why this simple augmentation strategy, or put it another way, weighting input and features from the concept library, could help improve vision-language tasks. Is this augmentation enough?

**Questions:**

See the weakness section.

---

> ### Author Response · Authors · 2023-11-15
> **Response to Reviewer Eqqp**
>
> Dear Reviewer Eqqp,
>
> Thank you for your detailed comments and suggestions. We tried our best to address all the concerns and questions, and update the main paper and appendix in the new version. Please let us know if you have any further concerns or questions to discuss.
>
> Best,
>
> Paper 4148 Authors
>
> ---
>
> ### Vocabulary construction process
> - From our results in Tables 1 and 2 and our analysis in Sec 4.2 and 4.3, the dataset-specific variants DCI-A and DCI-LM achieve better results, but we would also like to highlight that: **even with the generic vision vocabulary, our training-free DCI can already achieve comparable results with SOTA that requires training, which has already validated the main message that current paradigm of training projection and adaptation layers are not necessary.**
> - We indeed included the details of the vocabulary construction in Sec. 3.3 and G.1. The generic vocabulary is obtained from existing computer vision benchmarks – VisualGenome and OpenImages. The DCI-A dataset-specific vocabulary is obtained from the answer vocabulary used in the open-ended setting for each dataset.
> - We indeed included detailed quantitative results and analysis in Sec. F.2 for ablation study on DCI-LM. We generally observe that DCI-LM helps to remove from the concept recognition results some too generic visual concepts that are not relevant to the question.
>
> ---
>
> ### Insights and limitations
> - We indeed provided insightful analysis in Sec. 4.4 and 4.5. In Sec. 4.4, we specifically study how the projection layers can be replaced with different training-free variants, and we find that most of the ability of the learned projection layers and the adaptation layers can be instantly
> obtained with the simple constructed projection layer, as long as we use concepts as the bridge. This observation is aligned with our first insight that concepts can be directly understood by the language model, as summarized in the Introduction.
> - In Sec 4.5, we find that the augmentation in the intermediate feature space helps the model attend more to extracted concepts that are relevant to the correct answer, which is aligned with our second insight that the feed-forward process is the other ignored complementary pathway for modality fusion.
> - In Sec. 3.2.2, we also show the insights on how the injection process is equivalent to a feed-forward network with constructed weights, which again shows that as long as we speak the “language” that PLMs understand, the training process can be removed.
> - To further understand the potential of the proposed method, we also provide Plug-and-play results, results using ViT pretrained on ImageNet instead of CLIP, and Fine-tuning results in Appendix B, D, and F.1. We find that for models that have already been trained, our model can still provide help. When there is finetuning involved, DCI is also a better strategy to fuse visual information compared to the existing approach. More importantly, when there is no high-quality vision-language contrastive model like CLIP available, we show that with a generic classifier, our DCI method can still achieve better results compared to the existing approach.
> - Our method certainly has its limitations, which we discuss in detail in Appendix E. Overall, the main aim of this paper is not to completely replace existing approaches but to open up new possibilities for future research on better designs of modality fusion and new paths towards more efficient and versatile utilization of PLMs.

---

### Official Review · Reviewer_HYCX · 2023-10-31

**Soundness:** 3 good
**Presentation:** 3 good
**Contribution:** 3 good
**Rating:** 6
**Confidence:** 3

**Summary:**

This paper introduces a novel approach for zero-shot video/visual question answering leveraging the capabilities of a Large Language Model. To achieve this, the authors translate relevant visual concepts into textual representations using a predefined vocabulary set. Subsequently, they introduce the Deep Concept Injection Module, which integrates these textual visual concepts into the input and feed-forward networks. The effectiveness of this method is validated through extensive experiments conducted on eight video question datasets.

**Strengths:**

1. This paper proposes to inject the textual visual input into the feed-forward network, with experimental results affirming the efficacy of this approach.
2. Extensive experiments on 8 benchmark video question answering datasets demonstrate the effectiveness of the proposed method.

**Weaknesses:**

1. As shown in Eqn. (8), the input features of feed-forward networks guide the aggregation of the output features of feed-forward network, which is hard to comprehend. More explanations are required.
2. The authors adopt PLM to extract the features of textual visual concepts. However, when the length of the textual visual concepts exceeds one, it would be better to elucidate the specific feature extraction process.
3. There exists ‘?’ and ‘-’ in Table 1. It would be better to explain the meaning of these quotes.
4. While the ablation studies regarding hyper-parameters are included in the appendix, it would be beneficial to mention the best hyper-parameter settings in the implementation details to make the paper more concrete.
5. This method demonstrates remarkable performance. Could the authors consider releasing the source code upon acceptance?

**Questions:**

Please refer to the weakness

---

> ### Author Response · Authors · 2023-11-15
> **Response to Reviewer HYCX**
>
> Dear Reviewer HYCX,
>
> Thank you for your detailed comments and suggestions. We appreciate your recognition of the soundness, contribution, and presentation of our work. We tried our best to address all the concerns and questions, and update the main paper and appendix in the new version. Please let us know if you have any further concerns or questions to discuss.
>
> Best,
>
> Paper 4148 Authors
>
> ---
>
>
> ### Explanation of Eq. 8
> - Thanks for the suggestion. Eq. 8 adds the representation of concepts weighted based on the conditional distribution from Eq. 5 to the representation of the current word being processed, in the output space of each feed-forward network. $\lambda$ controls the strength of the augmentation. We have added this explanation in the revised paper.
> - We indeed included the intuitive explanation of this process in the last paragraph of Sec 3.2.2:
> > Intuitively, as verified in Figure 4, the intermediate representation of “[mask]” could not be close to the answer “hat” but after adding the representation of observed concepts, the model can make the correct prediction.
>
>
> ---
>
> ### When the number of tokens is larger than one for one concept
> - We include this detail in Appendix G.1 due to space limit. We simply apply averaging over the tokens when the concept consists of multiple tokens.
>
> ---
>
> ### Explanation of “?” and “-”
> - Thanks for the suggestion. “-” means not applicable and “?” means unclear from the original paper.
> - We have included it in the revised version.
>
> ---
>
> ### Hyper-parameter setting
> Thanks for the suggestion. We have added it to the revised main paper.
>
> ---
>
> ### Code release
> - Thanks for the question. We indeed plan to, as we mentioned in Sec. G.1.

---

### Official Review · Reviewer_YWMx · 2023-11-01

**Soundness:** 3 good
**Presentation:** 3 good
**Contribution:** 3 good
**Rating:** 6
**Confidence:** 3

**Summary:**

The paper presents a feed forward method to inject text prompts from visual inputs, for visual understanding tasks using large language model. Three variants of the method are presented, which are tested on different visual QA and dialogue datasets. Comparisons show that even without fine tuning the LLM with visual inputs, the system performs comparable or slightly better than competing algorithms.

**Strengths:**

The strengths of the paper are as follows:

1. The method does not require any training, and works simply augmenting the forward path of LLM by injecting semantic visual concepts.
2. Experiments have been conducted on various QA datasets assessing the performance of the method. The method looks like performing at par or better than some of the prior methods which do joint visual and text encoder training. It beats Flamingo, a visual-text encoder comprehensively on visual QA.
3. The method can be used easily with different PLMs and paper shows application of the method in multi-modal dialogue system, where the method outputs look reasonable.

**Weaknesses:**

The weaknesses are as follows:

1. The paper describes two variants of the DCI method in Section 3.2.1 and Section 3.2.2, however the variations presented in the experiments are based on the vocabulary selection method (DCI, DCI-A, DCI-LM). The authors have not assessed the performances of the previously described variants.
2. In experiment section, the visual encoder has not been mentioned explicitly. Authors need to add that information in the tables as well as description.
3. The vocabulary addition is a major step of the algorithm. Certain details and variations in vocabulary are missing: 1. What is the total number of visual concepts taken in the experiments, 2. What is the difference in output vocabulary of DCI-LM vs other variants. In equation 10, how are authors going from output of LLM to visual concepts. Question by itself can generate very open ended responses from LLM. Similarly for DCI-A, what is the number of top frequent words taken as dictionary.
4. In Table 2, there is not much difference between the results of BLIP-2 and proposed DCI variants. Authors have not explained the reason behind no significant change in output accuracy wrt original BLIP-2.
5. Examples of multimodal dialgoue systems show several images with named entities like Great Wall of China, orchid, etc. How is the current framework accounting for named entities in their method? Just visual input can potentially generate hallucinations in the LLM output. Authors have not explored the quality of the system in any details in the paper, hence the section does not add to the contribution of the paper.

**Questions:**

Questions to authors have been posted in weakness section.

**Details Of Ethics Concerns:**

Vocabulary generation is a major step in the algorithm. The source and potential biases of the vocabulary is not clear in the paper.

---

> ### Author Response · Authors · 2023-11-15
> **Response to Reviewer YWMx**
>
> Dear Reviewer YWMx,
>
> Thank you for your detailed comments and suggestions. We appreciate your recognition of the soundness, contribution, and presentation of our work. We tried our best to address all the concerns and questions, and update the main paper and appendix in the new version. Please let us know if you have any further concerns or questions to discuss.
>
> Best,
>
> Paper 4148 Authors
>
> ---
>
>
> ### Two Injection Mechanisms
> - As discussed in the introduction, Figures 1 and 3, Sec. 3.2, Sec. 3.2.1 and Sec. 3.2.2 are complimentary pathways that combine to form Deep Concept Injection.
> - We indeed included qualitative analysis in Figure 4 and Sec. 4.5 to analyze the effect of the constructed adaptation layers. We find that, with the help of the constructed adaptation layers, we successfully augment the intermediate feature to help the model attend more to extracted concepts relevant to the correct answer.
> - We also included quantitative analysis in the appendix due to space limit. In Sec. F.3, we quantitatively analyze the contribution of the two mechanisms and observe that combining the two injection mechanisms yields the best result.
>
> ---
>
> ### Visual encoders used
> - We indeed included the visual encoders used in the experiments in Sec. 4.1. As a summary, to facilitate fair comparisons, we use the same CLIP ViT-L/14 as FrozenBiLM when comparing with it, and we use the same Q-Former based on ViT-g as BLIP-2 when comparing with it.
> - We also included the information in the caption of Tables 1 and 2.
> - We are happy to indicate this further if the reviewer kindly provides more detailed suggestions.
>
> ---
>
> ### Vocabulary construction and ethic question
> - We indeed include the details of the vocabulary construction in Sec. 3.3 and G.1. We don’t expect additional bias to be introduced beyond that of the VisualGenome ontology.
> - We indeed included detailed quantitative results and analysis in Sec. F.2 due to the space limit. We generally observe that this process helps remove from the concept recognition results some too generic visual concepts that are irrelevant to the question. We will further add some qualitative examples of the difference between the retrieved concepts when the generic and narrowed vocabulary are used.
> - For DCI-LM, as shown in Eq. (10), we didn’t let the PLM decode in an open-ended manner. We use it to measure the relevance of each word in the generic vocabulary to the question.
> - Once we obtain the conditional probability given Eq. (10), as explained in the last sentence of page 6, we just select the most probable $n_C$ words as the concept vocabulary for retrieval and injection usage.
> - We didn’t change the size of the vocabulary for DCI-A. We just use it as it is, based on the setting of open-ended video/visual question answering. Therefore, it varies from dataset to dataset.
>
> ---
>
> ### Comparison with BLIP2
> - Thanks for the good question. As we mentioned in Sec G.2, all the hyper-parameters are tuned based on iVQA validation set to avoid fitting other datasets for zero-shot evaluation, which means that current hyper-parameters are not optimal for comparisons with BLIP-2.
> - Our results do not rely on any training, which is not expected to achieve significantly higher accuracy because of its great efficiency. The comparable results we have obtained are already quite surprising and validate the research hypothesis that the projection and adaptation layers are not necessary to be learned.
>
> ---
>
> ### Named entity
> - Thanks for the good question. Currently, we don’t directly handle named entities in our vocabulary, but this ability can be further integrated if we can also provide a list of named entities that we want the model to recognize.
> - For the Great Wall image, the recognized concepts include “china”, “fortification”, “tourism” etc. The PLM successfully inferred the most famous Great Wall based on these concepts.
> - For the orchid image, we respectfully disagree that “orchid” is a named entity. An "orchid," in general, is a type of flower and falls under a common noun. For this example, the recognized concepts directly contain the concept “orchid”.
> - We have also included more discussion about named entity in the revised Appendix H.

---

### Author Response · Authors · 2023-11-22
**[Updates] General Response**

We thank all the reviewers for their valuable comments.

We are encouraged that the reviewers appreciate our work because of its

    (1) good contribution to the community (YWMx,HYCX,Hqos), especially the intriguing design that constructs projection networks to replace fine-tuning (Hqos);
    (2) effectiveness, especially the strong performance achieved without any training (YWMx,HYCX,Eqqp,Hqos);
    (3) clear presentation (YWMx,HYCX,Hqos).

We would like to again highlight that our key insights in this work are that

    (1) it is not necessary to train these additional projection layers and adaptation layers (Sec 4.2, 4.4);
    (2) even if we have labeled data to fine-tune the model, our design is orders-of-magnitude more efficient with even better accuracy (Sec F.1).

We believe these insights, together with other findings provided in the paper, would
- **significantly improve the efficiency by providing a training-free solution**, and
- **open up new research on both rethinking existing designs and exploring more versatile algorithms along this line**.

We tried to address all the concerns and questions in detail, following the suggestions and comments of reviewers. We hope our responses addressed the concerns of reviewers. We are looking forward to discussing with reviewers on any further questions.

- [Update] We submitted a new revision annotating revised parts as blue to make it easier for reference.

---

### Meta-Review · Area_Chair_kCGX · 2023-12-05

**Metareview:**

The paper proposes a method to inject visual concepts into pretrained LLMs without training. It injects the most probable concepts as additional text inputs and augmentation in intermediate features. The proposed method achieves comparable or slightly better performance on 8 visual QA datasets.

Reviewers had raised concerns about the equations, lacking details and variations of the vocabulary, weak experimental results and limited novelty. The rebuttal did not well address the reviewers’ concerns. Two reviewers leaned towards rejection while two rated borderline accept scores. The AC agrees with the reviewers that the paper needs to improve the clarity of the method and the experimental ablations. Therefore, the AC recommends rejection.

**Justification For Why Not Higher Score:**

Explained in the metareview.

**Justification For Why Not Lower Score:**

N/A.

---

### Decision · Program_Chairs · 2024-01-16

Reject